# Antibiotic Resistance Profiles and ARG Detection from Isolated Bacteria in a Culture-Dependent Study at the Codfish Industry Level

**DOI:** 10.3390/foods12081699

**Published:** 2023-04-19

**Authors:** Gianluigi Ferri, Carlotta Lauteri, Mauro Scattolini, Alberto Vergara

**Affiliations:** 1Department of Veterinary Medicine, Post-Graduate Specialization School in Food Inspection “G. Tiecco”, University of Teramo, Strada Provinciale 18, 64100 Teramo, Italy; clauteri@unite.it (C.L.); avergara@unite.it (A.V.); 2Food and Health Veterinary Practitioner, 64015 Nereto, Italy; mauroscattolini58@gmail.com

**Keywords:** seafood industry, *Gadidae*, salted and seasoned products, soaked products, high pressure processing, antibiotic resistance, antibiotic resistance genes, one-health

## Abstract

The antibiotic resistance phenomenon horizontally involves numerous bacteria cultured from fresh or processed seafood matrix microbiomes. In this study, the identified bacteria from food-producing processes and industrial environments were screened for phenotypic and genotypic resistance determinants. A total of 684 bacterial strains [537 from processed codfish (*Gadus morhua* and *Gadus macrocephalus*) products as salted and seasoned and soaked and 147 from environmental samples] were isolated. Antibiotic susceptibility tests showed resistance against tetracycline, oxacillin, and clindamycin in the *Staphylococcus* genus (both from food and environmental samples) and against beta-lactams (cefotaxime, carbapenems, etc.) and nitrofurans (nitrofurantoin) from *E. coli* and *Salmonella enterica* serovar. Enteritidis isolates. One-thousand and ten genetic determinants—tetracycline *tet*C (25.17%), *tet*K (21.06%), *tet*L (11.70%), clindamycin *erm*C (17.23%), *erm*B (7.60%), linezolid *cfr* (8.22%), *optr*A (3.62%), *poxt*A (2.05%), and oxacillin *mec*A (17.37%)—were amplified from Gram-positive resistant and phenotypically susceptible bacteria. Concerning Gram-negative bacteria, the beta-lactam-resistant genes (*bla*_TEM_, *bla*_CIT_, *bla*_CTX-M_, *bla*_IMP_, *bla*_KPC_, *bla*_OXA-48-like_) represented 57.30% of the amplified ARGs. This study found high antibiotic resistance genes in circulation in the fish food industry chain from the macro- to microenvironment. The obtained data confirmed the diffusion of the “*antibiotic resistance phenomenon*” and its repercussions on the One-health and food-producing systems.

## 1. Introduction

The increased failure of antibiotics for therapeutic purposes in human and animal medicines represents a crucial public health concern. Among bacterial pathogens, the zoonotic strains, both for fish and human species, have widely demonstrated (during their evolutive *iter*) consistent sanitary and economic implications in numerous countries. Relevant microbiological *noxae*, such as *Listeria monocytogenes*, *Salmonella* spp., *Escherichia coli*, and *Staphylococcus aureus*, acquired new resistance strategies against the commonly administrated molecules to treat their infections. The EFSA report denounced the critical increase of intoxication cases responsible for human death, and the notable numbers of panresistant nosocomial strains isolated from human hospital environments [1]. The strict correlation of resistance determinants between different bacteria (from food, animal, or human skin microbiomes) and antimicrobial treatment failures have led to a critical reduction of pharma efficacy. This condition has induced the survival of certain strains, which have also resulted in not being susceptible to modern, critically important antimicrobials [2]. Indeed, the wide administration or misuse of antimicrobial molecules in food-producing animals, and more specifically in conventional farming systems, have also contributed as further *stimulus* to the growth of this phenomenon [3]. The diffusion of the aquaculture zootechnic sector has led to high animal densities, and consequently, antibiotic therapies have become necessary to control potential infectious outbreaks. Indeed, due to the administrated antimicrobial molecules, many finfish species, i.e., *Salmo salar*, *Oreochromis niloticus*, *Sparus aurata*, *Dicentrarchus labrax*, and *Gadidae* Family (*Gadus macrocephalus* and *G. morhua,* etc.) have been widely farmed. Secondly, finfish farming has also had repercussions on the health of wild fish species, especially in the cases of improper wastewater management and improper administration of mariculture environments [4]. The lack of physical barriers, in combination with oceanic currents, involves a crucial role in the environmental diffusion of nucleotide-resistant forms (i.e., integrons, plasmids, etc.). In these conditions, microorganisms result as *gene drivers*. They were also defined as *mobilized reservoirs* by Loayza et al. [5] in the so-called antibiotic resistance genes’ (ARGs) *environmental life cycle*. A fascinating biochemical environmental aspect was explained by researchers demonstrating that high salt contents (i.e., marine waters, salted food matrices, etc.) are responsible for improved phenotypic expression to many antibiotics. This chemical language is translated into different membrane proteins, inducing structural modifications to the receptors (i.e., ion pumps) becoming not susceptible [6]. Indeed, ARGs were largely amplified from many fresh and salted fish products obtained from caught animals, i.e., *Salmo salar* and *Oreochromis niloticu*, and successively handled in many food production chains. This last consideration suggests that the microbiological impacts on food spoilage are strongly influenced by cross-contamination due to the improper application of hygienic measures (in agreement with EU Reg. No. 852/2004) [7]. The European Food Safety Authority [1] highlighted that food matrices and their relative microbiomes could represent potential sources for the horizontal transmission of ARGs to the final consumers [5,8,9]. Many authors, based on the biomolecular ARG trades between food matrices and human microbiota, performed PCR assays, amplifying numerous genetic determinants from many bacterial species isolated in fresh fillets (fish and mammalian species). Molecular tests were designed to discover nucleotide sequences, which codify resistance against the most frequently administrated antibiotic molecules in veterinary and human medicine (i.e., tetracyclines, beta-lactams, and quinolones) [10,11,12,13]. They discovered the phenotypic expression of many antibiotic resistance patterns against the veterinary legally permitted molecules and the so-called Critical Important Antimicrobials (CIA), whose usage is strongly indicated for humans only [2]. Among the different finfish species, *G. macrocephalus* and *G. morhua* are largely caught wild; however, the aquaculture systems, mainly located in Northern Europe (i.e., Norway, Scotland, etc.), involve a strategic role. Indeed, they have improved production volumes in an attempt to satisfy the high demand for animal-origin protein for human and animal feeding. More specifically, the *Gadidae* Family has specific nutritional characteristics: high protein and low-fat contents. Their processed products, known as salted and seasoned codfish in European and South American countries, can provide the necessary nutrients to the human diet [14,15]. These processed fish matrices were poorly screened for antibiotic susceptibility or ARG detection (with special regard to horizontal gene transmission). Therefore, the objective of this research study was to provide antibiotic resistance profiles as the phenotypic expression of resistance genes amplified from isolated bacterial strains. These were obtained from differently processed codfish products (belonging to the fish species *G. macrocephalus* and *G. morhua*) and the environment along an integrated industrial supply food chain. This biomolecular study wants to provide preliminary data regarding antibiotic susceptibility tests and ARG detection starting from a culture-dependent investigation of bacterial strains isolated in the codfish industry.

## 2. Materials and Methods

### 2.1. Samples Collection

#### 2.1.1. Food Matrices

A total of 450 finfish products belonging to the *Gadidae* Family were involved in the sampling activities. Half of them were *G. macrocephalus* caught in the FAO zone 67 (Northern Pacific Ocean) and the others were *G. morhua* caught in the FAO zone 27 subarea IIa (Norway Sea, Atlantic Ocean). In accordance with European legal requirements EU Reg No. 1276/2011 and EU Reg. 625/2017, the primary producers immediately deheated and eviscerated all fish after catching. This last measure is considered necessary to reduce the migration of *Anisakis* spp. larval forms (L3 stage) from fish intestine to muscle tissues. These fish were successively salted and seasoned. After these steps, samples were imported by an industrial producer and were sectioned, producing fillets (muscle and skin tissues) characterized by an average weight of 400 ± 20 g/fillet. In accordance with consumer requests, many fish industries apply innovative technological processes, i.e., the soaking process and the subsequent exposure to High-Pressure Procedures (HPPs), to provide *ready-to-cook* and microbiologically safe food matrices [16]. These measures have resulted in being able to prolong the shelf life of products [17]. More specifically, 225 specimens collected from the two screened fish species were composed of three groups formed by 75 salted and seasoned (SD), 75 soaked (SP) and 75 HPP-treated samples. In agreement with international standardized methods, the soaking process was successively performed by the industrial fish producer in steel tanks using refrigerated waters +3 °C for 5–6 days. During this step, water was changed three times per day at regular intervals, as reported by Rode and Rotabaak [15]. After the soaking process, the same products were microbiologically stabilized by exposing them to the HPP technology. It was performed using the QFP 2L-700 system manufactured by Avure Technologies^TM^—HPP (Middletown, OH, USA) at the machine setting of 600 MPa for 5 min due to its high bactericidal impact, as previously described [15].

#### 2.1.2. Process Samples

One hundred processing samples (as food operator hands and industrial surface swabs) were also included in this study. More specifically, 50 specimens were collected from hand and 50 from surfaces, which usually were in contact with fish food products. Samples collections were performed after cleaning, washing, and disinfection procedures, as mentioned in the HACCP industrial document. The swabbing procedures were applied for their microbiological screenings (Sterile Swab Stick—Genorex Medsolutions, Suzhou City, Jiangsu Province, China) covering a total area of 100 cm^2^, in accordance with the UNI EN ISO 18593:2018. After collection, all samples were transported under refrigerated conditions and processed until 8 h from receipting.

### 2.2. Qualitative Microbiological Screenings

#### 2.2.1. Food Matrices

From each fish fillet sample type, two aliquots of 25 g and one of 10 g were sterilely collected using mono-usage scalpels (Monopec Scalpels, Thermo Fisher Scientific^TM^, Waltham, MA, USA). They were introduced in sterile stomacher bags (BagMixer^®^, Interscience, Puycapel, Cantal, France) in accordance with ISO 6887-3:2017. Thus, from each fish fillet, a total of three muscle tissue parts were collected for *Enterobacteriaceae*, *Listeria monocytogenes*, *Pseudomonas* spp., *Staphylococcus* spp., and *Vibrio* spp. qualitative detection. The first 25 g were diluted with 225 mL of Buffered Peptone Water (BPW) for *Enterobacteriaceae*, *Enterococcus* spp., and *Pseudomonas* spp. and the others in 225 mL Alkaline Peptone Water (APW) for halotolerant strains (*Staphylococcus* spp. and *Vibrio* spp.). The last 10 g were diluted with 10 mL of supplemented (Half Fraser Supplement, ThermoFisher Scientific, Waltham, MA, USA) and Half Fraser Broth (HFB) (Half Fraser Broth, ThermoFisher Scientific, Waltham, MA, USA) for *Listeria monocytogenes* selective culturing. All performed dilutions were performed following the ratio 1:10. After these steps, samples were stomached for 60 s and incubated at 37 °C for 24–36 h. From each broth, an aliquot was directly plated onto specific and supplemented agar media and successively incubated (in accordance with the culturing procedures mentioned in the referenced International Standards reported in Table 1). After this step, on the respective selective media, colonies’ morphological aspects were also considered as preliminary factors for identification, in agreement with the procedures (Table 1), but also supported by the following reported procedures.

#### 2.2.2. Environmental and Personnel Sample Collections

The swabbing method was used through sterile swabs and delimitators for 100 cm^2^ surfaces (Syntesys Disponsable Labware, Padova, Italy) and swabbing was performed directly from food operators’ hands. The collected samples were directly plated onto specific agar culture media for a qualitative investigation of *Staphylococcus* spp. and *Enterobacteriaceae*. These microbiological parameters were selected as quality and efficacy indicators of industrial hygienic preoperative sanitary measures (applied by the food operators along the productive lines). However, the other mentioned microorganisms (i.e., *Listeria monocytogenes*) were also considered. All strains were isolated according to the international standardized methods, as mentioned in detail in Table 1. After plating, specimens were incubated at 37 °C for 24–36 h.

### 2.3. Bacterial Identification and Antibiotic Susceptibility Tests (ASTs)

The bacterial identification and AST evaluations were performed using the biochemical automated method, VITEK^®^ 2 system, following the manufacturer’s procedures (bioMérieux, Paris, France). Gram-negative and positive strains were identified, obtaining results after incubation periods of 8 h from sample processing. Specific cards, VITEK^®^ ID-GN and VITEK^®^ ID-GP (VITEK^®^ 2 system, bioMérieux, Paris, France), were loaded and processed following the producer’s instructions. The biochemical assays were successively confirmed through the mass spectrometry MALDI-TOF (Matrix Assisted Laser Desorption Ionization—Time of Flight). The ASTs were also performed with the VITEK^®^ 2 system (bioMérieux, Paris, France) as an automated device providing results between 22–24 h from samples’ loading. Parallelly to the antibiogram assays, following the same protocol, the Minimum Inhibitory Concentration (MIC) values were calculated for each resistant bacterial strain. Bacterial suspensions, with a final density of 0.5 McFarland standard, were realized. A final volume of 280 µL for Gram-negative strains and 145 µL for Gram-positive bacteria were collected from the suspensions and added to 3 mL of VITEK 0.45% saline solution. Gram-negative ASTs were performed using the card named AST-N379, which tested 16 antibiotic molecules (amikacin, amoxicillin/clavulanic acid, cefepime, cefotaxime, ceftazidime, ciprofloxacin, colistin, ertapenem, ESBL, fosfomycin, gentamycin, imipenem, meropenem, nitrofurantoin, and sulfamethoxazole) belonging to the most frequently administrated antibiotics in veterinary medicine and part of the CIA lists.

The AST-P658 and AST-P659 were used for Gram-positive strains, including 26 antibiotics (amoxicillin/clavulanic acid, ampicillin, ciprofloxacin, daptomycin, gentamicin, imipenem, kanamycin, levofloxacin, linezolid, nitrofurantoin, quinupristin/dalfopristin, streptomycin, teicoplanin, tigecycline, trimethoprim, trimethoprim/sulfamethoxazole, vancomycin, benzylpenicillin, cefoxitin, cefazoline, clindamycin, erythromycin, mupirocin, oxacillin, rifampicin, and tetracycline). The obtained susceptibility results were elaborated in accordance with the Clinical & Laboratory Standards Institute (CLSI) breakpoints determined as relevant for humans [24]. Bacterial isolates that resulted resistant to three or more different antibiotics were classified as multidrug resistant (MDR), as previously reported by Magiorakos et al. [25].

#### MRS and MSS *Staphylococcus* spp.

Before PCR assays, all identified Staphylococci were also tested using the CHROMID^®^ MRSA—bioMérieux—Culture Media (Paris, France) as possible MRS strains. After this first analysis and ASTs, all of them were also bimolecularly screened for detection of the so-called *mec*A gene, which is responsible for methicillin and oxacillin resistance patterns. The phenotypic and the genotypic confirmation permitted classifying Staphylococci as methicillin-resistant (MRS) or methicillin-susceptible (MSS). Specific primers, designed by McClure et al. [26], and their respective PCR setting parameters were performed.

### 2.4. Biomolecular Assays

#### Bacterial DNA Extraction and ARG Screening

A culture-dependent approach was performed both for food matrices and process samples. The DNA extracts were obtained using the High Pure PCR Template Preparation Kit (Roche^®^, Indianapolis, IN, USA), obtaining final volumes of 100 μL. These aliquots were stored at −20 °C until their biomolecular screenings. The PCR assays, performed as uniplex or multiplex reactions, were realized introducing specific primers in the reaction volumes, as reported in Table 2. The screened ARGs included the veterinary legally permitted molecules as beta-lactams (*bla*_TEM_, *bla*_CTX-M_, *bla*_CIT_,) and tetracycline (*tet*A, *tet*C, *tet*M, *tet*K, *tet*L), and the CIAs as vancomycin (*van*A, *van*B, *van*C1, *van*C2, *van*D, *van*M, *van*N), carbapenems (*bla*_IMP_, *bla*_NDM_, *bla*_OXA-48-like_, *bla*_KPC_), etc. [2]. A list of all tested genes is included in Table 2. The thermocycler and annealing settings were in accordance with their respective reference indications.

The PCR reactions were performed in a total volume of 25 μL/reaction using the Green Master Mix Promega^®^ (Milano, Italy) and then 1 μL of the extracted DNA was added to the respective tube. The amplification conditions were performed following the procedures indicated in the respective references, which are listed in Table 1. This step was followed by electrophoresis, which was realized by loading the obtained amplicons on the agarose gels (1.0–1.5% depending on band sizes, setting at 80 Volt for 30 min). The nitid bands were compared with specific DNA ladders (Genetics, FastGene 100 bp or 100–1000 bp DNA Marker, Düren, Germany). The suspected positive samples were purified using the Qiagen QIAquick^®^ PCR Purification Kit (Hilden, Germany). The obtained oligonucleotide specimens were successively sequenced by BioFab Research (Rome, Italy) through the Sanger method. The nucleotide similarities analyses were performed using the BLASTN system (http://www.ncbi.nlm.nih.gov/genbank/index.html) (accessed on 23 July 2022).

### 2.5. Statistical Analysis

The statistical analysis was performed using the SPSS 20.0 (SPSS, Chicago, IL, USA). The *t* test was applied to compare differences between resistant isolated bacteria both from food matrices and the environment. The same test was also used to analyze ARG frequencies. Results were considered statistically significant in the case of a *p* value < 0.05. For all prevalence percentage values, confidence intervals (CI 95%) were calculated when applicable.

## 3. Results

### 3.1. Bacterial Identification

In this biomolecular study, a total of 684 bacterial strains were isolated from 450 fish food matrices and 100 process samples (including food operator hands and industrial surfaces). Four hundred and twenty-five Gram-positive (62.13%) and 259 Gram-negative (37.87%) strains were identified from all specimens, as illustrated in Table 3.

*Staphylococcus* genus was the most identified in both specimens. More specifically, 73.17% (95% CI: 68.96–77.38%) or 311/425 Gram-positive isolates belonged to the mentioned genus. Two-hundred forty-seven out of 311 (79.42% 95% CI: 74.93–83.91%) were isolated from food matrices, while 64/311 (20.57% 95% CI: 16.08–25.06%) were isolated from process specimens (See Table 4).

*Enterococcus* spp. was also widely identified, representing 16.0% or 68/425 (95% CI: 12.52–19.48%) of Gram-positive bacteria. Forty-four out of 68 isolates were *Enterococcus faecalis,* representing 64.7% (95% CI: 53.35–76.05%), followed by *Enterococcus durans* 17/68 or 25.0% (95% CI: 14.71–35.29%), and, finally, 7/68 *Enterococcus faecium*, representing 10.29% (95% CI: 3.07–17.51%). Among the Gram-positive bacteria, 35/425 (8.23% 95% CI: 5.62–10.84%) were *Kocuria* spp., and specifically, 24/35 were classified as *Kocuria kristinae* (68.57% 95% CI: 53.19–83.95%), and 11/35 as *Kocuria varians* (31.42% 95% CI: 16.05–46.79%). The least-identified genus was *Micrococcus* spp., with 11/425 (2.58% 95% CI: 1.08–4.08%). Most of these microorganisms were isolated from SD and SP products. Among the isolated Gram-negative bacteria, two bacterial species (reported in the EU Reg. No. 2073/2005) were identified, 18 *Escherichia coli* and two *Cronobacter sakazakii*, and one *Salmonella enterica* serovar Enteritidis from food matrices. Further detailed information is reported in Table 5.

The most representative Gram-negative microorganisms were *Acinetobacter lwofii* 62/259 (23.93% 95% CI: 18.74–29.12%), *S. paucimobilis* 50/259 (19.30% 95% CI: 14.50–24.10%), *Pseudomonas luteola* 41/259 (15.83% 95% CI: 20.27–11.39%), *S. fonticola* 37/259 (14.28% 95% CI: 10.02–18.54%), and *P. fluorescens* 27/259 (10.42% 95% CI: 6.70–14.14%).

### 3.2. AST Results

Regarding the AST results, wide phenotypic resistances were observed in the Gram-positive isolates against tetracycline, 312/425 or 73.41% (95% CI: 69.21–77.61%), and lincomycin (clindamycin), 201/425, of which 47.29% (95% CI: 42.55–52.03%) were resistant against other antibiotic classes, as shown in Table 6 and Table 7. One-hundred ninety-nine out of 311 bacteria (63.98% 95% CI: 58.65–69.31%), belonging to the *Staphylococcus* genus, were also resistant to oxacillin. Additionally, the negative coagulase *Staphylococcus* spp. resulted in being resistant to linezolid (13.18% 95% CI: 9.42–16.94%), oxacillin (63.98% 95% CI: 58.65–69.31%), and vancomycin (14.79% 95% CI: 10.85–18.73%). The resistance patterns against vancomycin and linezolid were widely observed in *Enterococcus* spp., at the rate of 22/68 32.35% (95% CI: 21.23–43.47%) and 33/68 48.52% (95% CI: 37.34–59.70%), respectively. Further detailed information regarding AST results is reported in Table 6.

Among the 425 Gram-positive strains, 230 bacteria, mostly isolated from SD and SP products before HPP treatment, resulted resistant to at least three antibiotic molecules belonging to different classes (beta-lactams, glycopeptide, lincomycin, oxazolidinone, and tetracycline). For this reason, these strains were classified as multidrug resistant (MDR). Specifically, 192/311 Staphylococci isolates (61.73% 95% CI: 56.33–67.13%) and 38/68 Enterococci isolates (55.88% 95% CI: 44.08–67.68%) resulted MDR, as indicated in Table 6 and Table 7. Six out of 46 isolated *Kocuria* spp. strains (13.04% 95% CI: 3.31–22.77%) were resistant only to tetracyclines.

More specifically, tetracyclines showed higher phenotypic resistance patterns than the other tested molecules among the *Staphylococcus* genus: 76.47% (95% CI: 46.64–92.88%) of *S. aureus*, 82.95% (95% CI: 75.09–90.81%) of *S. sciuri*, 87.50% (95% CI: 80.60–87.57%) of *S. lentus*, and 89.33% (95% CI: 82.35–96.31%) of *S. saprophyticus* strains. A similar result was observed among the *Enterococcus faecalis* genus [79.54% (95% CI: 67.62–91.46%)] (MIC values are illustrated in Table 6).

The 259 Gram-negative bacteria isolated from food matrices and process specimens mostly showed resistance against the cefotaxime molecule (beta-lactam class): 31/259 isolates corresponding to 11.96% (95% CI: 8.01–15.91%), as illustrated in Table 7.

Four out of 41 *P. luteola* (9.75% 95% CI: 0.75–18.75%) were classified as MDR due to their phenotypic resistance patterns against beta-lactams, carbapenem, and nitrofuran. Finally, only 2/18 *E. coli* strains (11.11% 95% CI: 0.37–15.89%) resulted MDR, showing similar resistance results to the above-mentioned antibiotic classes. Ten out of 50 *Sphingomonas paucimobilis* isolates (20.00% 95% CI: 8.92–31.08%) were not susceptible to the cefotaxime molecule. Gram-negative MDR bacteria were mostly isolated from salted and seasoned products.

### 3.3. ARG Detection

Among Gram-positive strains, and more specifically the *Staphylococcus* spp., tetracycline ARGs (*tet*C, *tet*K, *tet*L), clindamycin (*erm*C), and oxacillin (*mec*A) were widely amplified by the biomolecular assays from food and process specimens. At same time, the genotypic screening identified numerous unencoded ARGs to other antibiotic classes, i.e., aminoglycosides, beta-lactams, carbapenems, and sulfonamides. Concerning MRS or MSS strains, 11/51 or 21.56% (95% CI: 10.28–32.84%) bacteria, which phenotypically resulted susceptible to oxacillin, also harbored the *mec*A gene. Comparing the *mec*A gene amplification, discovered from Staphylococci isolated in food matrices and environmental specimens, there was a statistical difference with a *p*-value < 0.001. As previously observed, *Enterococcus* spp. also showed ARGs against tetracycline, clindamycin, and linezolid antibiotic molecules. From a statistical perspective, the *Staphylococcus* genus harbored numerous ARGs presenting a statistical difference if compared with *Enterococcus* spp. (*p*-value: 0.001), as illustrated in Figure 1.

The corresponding genetic determinants were identified in all phenotypically resistant isolates. More specifically, the obtained AST data can be considered the translation results of specific ARGs. Among Gram-positive strains, *Staphylococcus* and *Enterococcus* genera were the prevalent microbial populations, as reported above, and tetracycline ARGs were largely amplified. These patterns were supported by the biomolecular amplification of different gene determinants, i.e., *tet*C, *tet*K, and *tet*L (See Figure 2).

Clindamycin resistance was also genotypically based on *erm*B and *erm*C detections. Concerning Gram-negative resistant bacteria, *Pseudomonas* spp. and *E. coli* strains mostly covered a driver role for beta-lactams (*bla*_TEM_, *bla*_CIT_, and *bla*_CTX-M_), carbapenems (*bla*_IMP_, *bla*_KPC_, *bla*_OXA-48-like_), and nitrofurantoin (*nfs*A and *nfs*B), which resulted statistically different from the other bacterial species *Acinetobacter* spp., *Sphingomonas* spp., and *Serratia* spp. with a *p*-value < 0.001 (See Figure 3).

Among Gram-negative DNA, specific ARGs, belonging to the beta-lactam antibiotic class, were mainly amplified: *bla*_TEM_ (4.25% 95% CI: 3.20–5.30%), *bla*_CIT_ (1.84% 95% CI: 1.13–2.55), *bla*_CTX-M_ (2.05% 95% CI: 1.31–2.79%), *bla*_IMP_ (1.49% 95% CI: 0.86–2.12%), *bla*_KPC_ (1.49% 95% CI: 0.86–2.12%), and *bla*_OXA-48-like_ (0.42% 95% CI: 0.09–0.75%). A complete scheme representing ARG distributions are reported in Table 8.

### 3.4. Statistical Analysis

From a macro perspective, the studied identified bacteria were mostly isolated from SD and SP products from both fish species. Significant differences were observed between SD and SP bacterial amounts (*p*-values < 0.0001) and between SP and HPP-treated products (*p*-value: 0.002). At the same time, there was also a significant difference comparing the number of microorganisms isolated from food operator hands and surfaces (*p*-value < 0.001). Concerning the AST results, the difference between Gram-positive and negative bacteria (isolated from foodstuffs) was statistically significant (*p*-value: 0.002).

From a biomolecular point of view, the statistical analysis also showed a significant difference in the ARG distributions between food matrices and process specimens (*p*-value < 0.001). Focusing on the food-isolated-resistant strains, the gene amplification amounts between SD, SP, and HPP-treated samples (both screened fish species) were statistically significant with *p*-values < 0.0001. More specifically, tetracycline genes (*tet*C and *tet*K), amplified from SD and SP in *Staphylococcus* spp., resulted statistically different (*p*-value < 0.0001). A similar result was observed for clindamycin ARGs [*erm*B (*p*-value: 0.001) and *erm*C (*p*-value < 0.001) genes] and oxacillin [*mec*A (*p*-value < 0.001)] between SD and SP specimens. *Bla*_TEM_ (beta-lactams), *bla*_IMP_, and *bla*_OXA-48-like_ (carbapenems) were differently amplified between SD and SP samples in both studied fish species (*p*-values: 0.001). More specifically, *bla*_TEM_ and *bla*_IMP_ presented statistically significant differences with *p*-values < 0.001 between SD and SP specimens. Based on the same comparison, *bla*_OXA-48-like_ showed a *p*-value of 0.015.

## 4. Discussion

This study performed a culture-dependent and biomolecular investigation focused on the AMR phenomenon and, more specifically, on the circulation of ARGs at the fish food industry level. These screenings involved the *Gadidae* Family, specifically, the *G. morhua* and *G. macrocephalus* finfish species, which are normally used for salted and seasoned codfish production [17]. All analytical steps were also organized and performed following the industrial processes, starting from raw and processed fish food matrices, including operator hands and industrial surfaces (the environmental variable). The ASTs were designed adapting to the screened industrial productive system or line peculiarities. The screened gene targets, for molecular biology assays, included legally permitted antibiotics for veterinary medicine (especially in the aquaculture zootechnic sector) and the so-called Critical Important Antimicrobials (CIA), the usage of which has been strongly recommended for humans only [2].

Gram-positive strains were mostly identified both from food matrices and process specimens, i.e., food operator hands. Among the halotolerant bacteria, the *Staphylococcus* spp., positive and negative coagulase species, was predominant, with 73.17% (95% CI: 68.96–77.38%) of the identified bacteria. These data were in line with previous studies on microbial communities (pathogens and spoilage bacterial species) that studied salted and seasoned products (belonging to the *Gadidae* Family) [14,40,41]. Among the *food hygiene criteria* (reported in the EU Reg. No. 2073/2005), 51 *S. aureus*, 18 *E. coli*, two *C. sakazakii*, and one *Salmonella* serovar. Enteritidis isolates were also identified. Their detection can be considered as evidence of human contamination that may possibly be explained by the improper application of the so-called “*Good Hygiene Practices*” (GHP) (reported in the EU Reg. No. 852/2004) [5]. Commensal Gram-negative bacterial species were widely discovered (as illustrated in Table 5), in line with previous microflora investigations [14,42]. These findings are explained by the food microenvironmental conditions (aw and pH values) that select and improve commensal bacteria multiplication; indeed, the described and discovered microbiota results are invariable and in line with previous studies, as reported by Rodrigues et al. [14] and Helsens et al. [43].

The AST results, involving all isolated bacteria, produced original preliminary data from differently processed fish products and environmental specimens, as schematically reported in Table 6 and Table 7. The most frequently discovered resistances, in Gram-positive bacteria (Staphylococci), resulted against the following antibiotic molecules: tetracycline and clindamycin, both from food matrices and food operator hands (as previously described in the Results section). Similar patterns were also observed by other authors who conducted microbiological surveys on salted or salted and fermented fish products [14,43,44]. The resistance data, observed in negative coagulase Staphylococci reported in the AST Results section, have also been largely observed in nosocomial isolates, especially in MRS strains [45]. Their potential role as drivers, due to their harboring of ARGs from food matrices to the final consumers’ microbiome has been largely confirmed, as recently reported by Timmermans et al. [46]. Gene cassettes, responsible for the genetic transmission of genetic determinants, were demonstrated to be involved in the spreading of genotypic resistance to tetracyclines, clindamycin, and oxacillin [44,45].

*Enterococcus* spp. also presented wide resistance to the linezolid molecule, which was added by the EMA in the CIA lists, and for this reason, it is not indicated for therapeutic purposes in veterinary medicine. Similar AST results were also observed in other Enterococci strains isolated from wild mammalian feces [47]. These findings can be explained by the high impacts of anthropic activities and their ecological repercussions on humans and wildlife. These observations have been explained by the European Agencies, focusing attention on the chemical pressures caused by improper antibiotic administration or misuse [1].

Among Gram-negative strains, *E. coli* resulted significantly resistant to the beta-lactam antibiotic class (especially against carbapenems and cefotaxime molecules) and nitrofurans (nitrofurantoin). Similar phenotypic patterns were also described in other salted finfish matrices, where the phylogenetic analysis of amplified ARGs (described in the following paragraph) confirmed the evolutionary human origins [5,40,41]. The obtained AST data were mostly discovered from food matrix specimens. Indeed, a significant difference was observed (*p*-value: 0.001) between product and food operator hand isolates.

Regarding ARG circulation, the biomolecular screenings involved all isolated bacterial strains. All PCR reactions were performed following the flow chart production, following previous affirmations mentioned in the Results section.

The ARG screenings revealed a high circulation, amplifying a total of 1.410 oligonucleotide determinants; more specifically: 11,015/1410 (78.37% 95% CI: 76.22–80.52%) were detected from phenotypically resistant strains supporting the genetic basis of the AST results and the remaining 395/1410 (28.01% 95% CI: 25.67–30.35%) from susceptible strains. This evidence enforced the scientific hypothesis concerning the critical role of many bacteria (including pathogen and commensal ones) as environmental reservoirs, confirming the scientific *alert* announced by the European Food Safety Agency. The scientific concern was the critical role of the susceptible bacterial reservoirs indicated as “*genetic environmental resistance forms*”. This last-mentioned role was largely documented from commensal strains [1,2].

Concerning ARG distributions, *tet*C (25.17% 95% CI: 22.91–27.43%), *tet*K (21.06% 95% CI: 18.93–23.19%), and *tet*L (11.70% 95% CI: 10.03–13.37%) for tetracycline and *erm*C (17.23% 95% CI: 15.26–19.20%) and *erm*B (7.60% 95% CI: 6.22–8.98%) for clindamycin were amplified, both from products and food operator hands (mainly from Gram-positive DNA). Similar detections were also observed in previous studies conducted on salted fish products, demonstrating that simultaneous resistance to tetracyclines and clindamycin may possibly be associated with gene cassettes [48,49].

These data agreed with the results shown by the isolated strains from salted and fermented fish products by Majumdaret and Gupta [50]. These findings can be explained by different macro- and microenvironmental conditions, which provided fundamental aspects facilitating so-called horizontal gene transmission. The first consideration is linked to the food specimen characteristics; salted and seasoned products are usually handled by different food operators from fishing (as primary products) to their processing at the industrial level. From a technological perspective, the salting process generally induces bacterial lysis and nucleotide structural alterations. However, halotolerant species (i.e., *Staphylococcus* spp., *Vibrio* spp.) can survive by adapting their osmotic *equilibrium* to the low aw values. This extracellular *stimulus* acts as an inductor for ARG trades [6]. Their detections cannot exclusively represent an improper GHP application. Indeed, *Staphylococcus* spp., which have frequently harbored these *tet* and *erm* genes, are part of the normal microbiome identified from salted and seasoned codfish or other salted or fermented fish species [14,29,43,51]. Secondly, horizontal ARG transmission has a key role in their spreading among organic substrates [1]. The genomic combination of resistance against tetracycline and clindamycin, among *Staphylococcus* spp., is usually related to possible gene cassettes, as first observed by Strommenger et al. [52].

In this study, 392/684, or 57.30%, (95% CI: 53.60–61.00%) bacterial strains harbored numerous ARGs (*bla*_TEM_, *bla*_CIT_, *bla*_CTX-M_, *bla*_KPC_, *bla*_IMP_, *bla*_OXA-48-like_, *tet*C, *tet*K, *tet*L, *erm*B, *erm*C, *sul*2, and *sul*3) against specific antibiotic classes that were not phenotypically expressed (See Table 8). Indeed, the wide detection of ARGs is not always associated with their encoding and its consequential transcriptions [29]. The suggested scientific hypothesis was that these commensal or pathogenic strains were involved as microenvironmental reservoir forms of resistance, in agreement with previous studies [29,51,53]. The genetic expressions depend on multiple factors, and results are strongly influenced by environmental stressors, which have resulted to be inductors of specific forms of resistance. An emblematic example was represented by the sodium-chloride efflux pumps encoding (i.e., salted foodstuffs, high-content human diets, etc.), which was demonstrated to be altered, as reported by genomic studies. This condition has led to the evidence that high salt contents can modify bacterial susceptibility to antibiotic molecules. This process was observed in the bacterial cross-protection against antibiotics determined by the increased *AcrAB-TolC* efflux pump expression levels. This physio-pathological pathway has conferred resistance to those molecules, whose pharma-dynamical action involves ion transmembrane protein structures. This aspect was experimentally observed by Zhu and Dai [6] in *E. coli* strains, which presented low stress-induced tetracycline and chloramphenicol susceptibility after exposure to high salt contents. The strong connections that emerged between cellular physiology (both intra and extra) and the environment further clarified consistent repercussions on the AMR phenomenon influencing antibiotic therapeutic efficacies [2]. The cytological interactions between human and food microbiomes may also be involved in the horizontal transmission of resistant forms involving pathogenic or commensal strains [53,54,55]. This condition represents one of the crucial aspects of the real complexity and pleomorphism of AMR concern.

Parallelly to the proper GHP applications, food industries have introduced new food technologies, i.e., the HPP for seafood products’ shelf-life prolongation. This last method (HPP) has determined consistent CFU/g reductions in processed fish products [17]. In this study, a significant difference between HPP-untreated and HPP-treated products was observed (*p*-values: <0.001). This technology demonstrated high efficacy against both Gram-negative and Gram-positive bacteria. Indeed, the highly selective culturing procedures, involved in this qualitative study did not identify pathogens or commensal strains. The obtained results were in line with the scientific findings reported by Arnaud et al. [16]. Furthermore, the DNA biomolecular screenings, performed on isolates identified from HPP products, did not amplify ARGs. A possible explanation was first proposed by Oliveira et al. [17] and successively confirmed by Rode and Rotabaak [15], regarding the HPP denaturation effects on the hydrogen bonds between DNA strands and on covalent strands between nucleotides.

This research, in consideration of EFSA report [1], highlighted as spoilage or pathogenic bacteria, isolated from food matrices or the environment, can harbor different ARGs, reinforcing the AMR phenomenon. Legally permitted GHP measures (EU Reg. No. 852/2004) in combination with innovative technologies (i.e., HPP) could provide safe foods. However, it is necessary that a significant reduction in antibiotic usage as the main selective pressure be considered [2]. Due to the complexity of the AMR phenomenon, this study wants to provide a preliminary investigation combining industrial food processes with ARG diffusion, proposing an environmental perspective from the macro- to the microworlds.

## 5. Conclusions

All performed PCR assays amplified ARGs from both food matrices and environmental cultured bacteria. Of the 425 Gram-positive strains (with special regard to the *Staphylococcus* genus), tetracycline (*tet*C; *tet*K; *tet*L), clindamycin (*erm*B and *erm*C), and oxacillin (*mec*A) resistance genes were largely discovered. Beta-lactams (*bla*_TEM_, *bla*_CIT_, *bla*_CTX-M_, *bla*_IMP_, *bla*_KPC_, *bla*_OXA-48-like_) and nitrofurantoin (*nfs*A and *nfs*B) genes were detected from 259 Gram-negative ones. Phenotypically resistant strains were mostly isolated from salted and soaked products, while the HPP food technology confirmed its sanitary application, highlighting its bactericidal effect and the ability to modify nucleotide macromolecular structures. The amplification of any genetic determinants, i.e., *cfr*, *optr*A, *poxt*A (carbapenem molecule), and *van*D (vancomycin), belonging to the so-called CIAs, provides an ecological perspective concerning ARG diffusion in the screened seafood industry. The obtained results further highlight the complexity of the AMR phenomenon and the importance of biomolecular surveillance in anthropized environments. For these reasons, deep attention to *Good Hygiene Practices* and high-sanitary levels applied during food technological processes will uncover fundamental roles in reducing possible cross-contaminations, in agreement with the new concepts introduced by the *Hazard Analysis and Risk-Based Preventive Controls* (FDA, 2018). Combining the approaches of food industries and the applied food microbiology sciences will provide more detailed explanations concerning cytological interactions and the inductive pathways involved in chemical signaling on extra- and intracellular structures.

## Figures and Tables

**Figure 1 foods-12-01699-f001:**
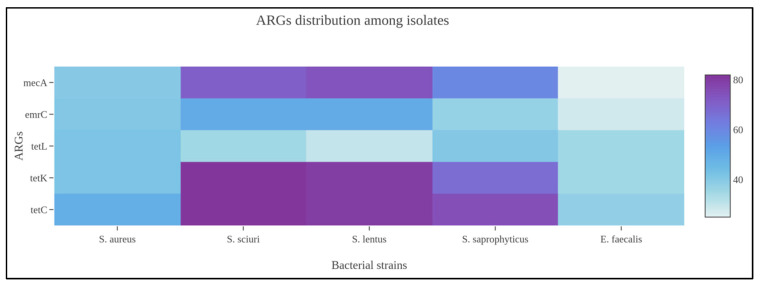
Heatmap representing ARG distribution among *Staphylococcus* spp. identified in the present scientific investigation.

**Figure 2 foods-12-01699-f002:**
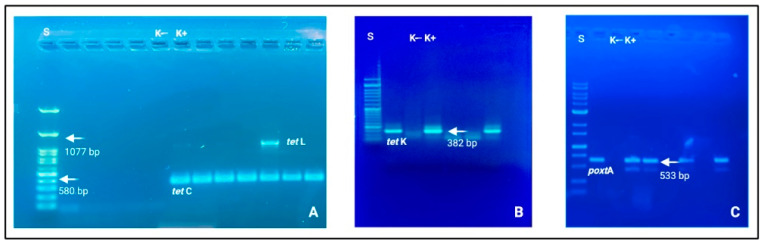
Electrophoresis gels (agarose 1.0 and 1.5%) in which it is possible to observe nitid positive amplicons and amplification results. (**A**) *tet*C (580 bp) and *tet*L (1077 bp); (**B**) *tet*K (382 bp); (**C**) *poxt*A (533 bp). Loading wells: S: DNA ladder 1 Kb (Genetic^®^ FastGene 100–10,000 bp DNA Marker, Düren, Germany); K−: negative control and K+: positive control.

**Figure 3 foods-12-01699-f003:**
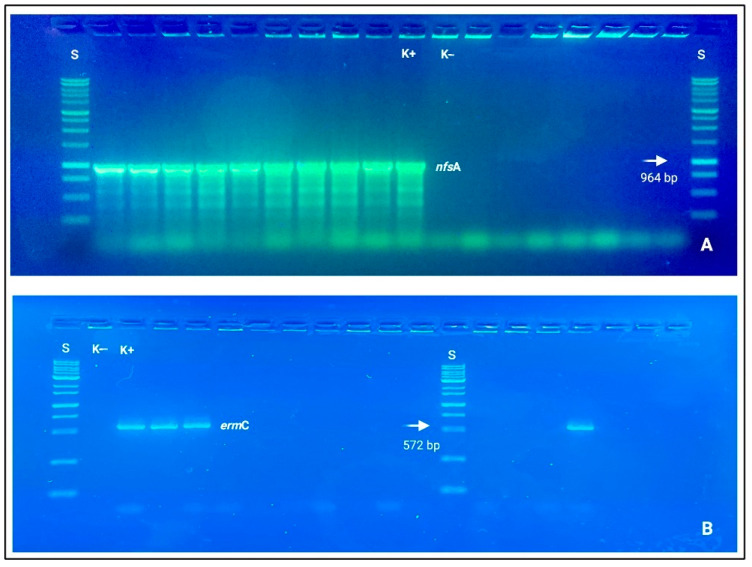
Electrophoresis gels (agarose 1.0%), in which it is possible to observe nitid positive amplicons and amplification results. (**A**) *nfs*A (964 bp); (**B**) *erm*C (572 bp). Loading wells: S: DNA ladder 1 Kb (Genetic^®^ FastGene 100–10,000 bp DNA Marker, Düren, Germany); K−: negative control and K+: positive control.

**Table 1 foods-12-01699-t001:** Selective culturing agar and broth media for bacterial pathogens (according to the EU Reg. 2073/2005) used in the present investigation.

Bacterial Strains	Culturing Broths	Selective Agar Media	Supplements	Standard Methods
*Listeria monocytogenes*	HFB	ALOA agar (OXOIDTM)	Listeria (OXOIDTM)	[18]
*Vibrio* spp.	APW	TCBS agar (OXOIDTM)	NaCl 20%	[19]
*Staphylococcus* spp.	APW	Baird–Parker agar (OXOIDTM)	R.P.F. (OXOIDTM)	[20]
*Enterococcus* spp.	BPW	Slanetz–Bartley agar		[21]
*Enterobacteriaceae*	BPW	Mac Conkey agar		[22]
*Pseudomonas* spp.	BPW	Pseudomonas agar	Pseudomonas C-F-C (OXOIDTM)	[23]

APW: Alkaline Peptone Water. BPW: Buffered Peptone Water. HFB: Half Fraser Broth.

**Table 2 foods-12-01699-t002:** Target genes used for ARG screening in the present study.

Categories	Target Genes	Primers	Amplicons (bp)	References
Aminoglycosides	*aph*A1	F: ATGGGCTCGCGATAATGTC R: CTCACCGAGGCAGTTCCAT	600	[27]
	*aph*A2	F: GATTGAACAAGATGGATTGC R: CCATGATGGATACTTTCTCG	347	
	*aad*B	F: GAGGAGTTGGACTATGGATT R: CTTCATCGGCATAGTAAAAG	208	
	*aac(3)*IV	F: TGCTGGTCCACAGCTCCTTC R: CGGATGCAGGAAGATCAA	653	
Beta-lactams	*bla* _TEM_	F: TCGCCGCATACACTATTCTCAGAATGA R: ACGCTCACCGGCTCCAGATTTAT	445	[28]
	*bla* _CTX-M_	F: GGGCTGAGATGGTGACAAAGAG R: CGTGCGAGTTCGATTTATTCAAC	876	[29]
	*bla* _CIT_	F: TGGCCAGAACTGACAGGCAAA R: TTTCTCCTGAACGTGGCTGGC	462	[30]
Carbapenem	*bla* _IMP_	F: GGAATAGAGTGGCTTAAYTCTC R: GGTTTAAYAAAACAACCACC	232	* [31]
	*bla* _OXA-48-like_	F: GCGTGGTTAAGGATGAACAC R: CATCAAGTTCAACCCAACCG	438	
	*bla* _NDM_	F: GGTTTGGCGATCTGGTTTTC R: CGGAATGGCTCATCACGATC	621	
	*bla* _KPC_	F: CGTCTAGTTCTGCTGTCTTG R: CTTGTCATCCTTGTTAGGCG	790	
Clindamycin	*erm*A	F: GTTCAAGAACAATCAATACAGGAG R: GGATCAGGAAAAGGACATTTTAC	421	* [32]
	*erm*B	F: CCGTTTACGAAATTGGAACAGGTAAAGGGC R: GAATCGAGACTTGAGTGTGC	359	
	*erm*C	F: GCTAATATTGTTTAAATCGTCAATTCC R: GGATCAGGAAAAGGACATTTTAC	572	
Linezolid	*cfr*	F: TGAAGTATAAAGCAGGTTGGGAGTC R: AACCATATAATTGACCACAAGCAGC	746	* [33]
	*optr*A	F: TACTTGATGAACCTACTAACCA R: CCTTGAACTACTGATTCTCGG	422	
	*poxt*A	F: AAAGCTACCCATAAAATATC R: TCATCAAGCTGTTCGAGTTC	533	
Nitrofurantoin	*nfs*A	F: CTGGCGCTTGCTCTGCTATC R: GCCCGCGTATCATACACTGG	964	[34]
	*nfs*B	F: ATCACCGTCTCGCTACTCAAC R: CGCGCCATTGATCATTGAGG	921	
Sulfamethoxazole	*sul*1	F: TGGTGACGGTGTTCGGCATTC R: GCGAGGGTTTCCGAGAAGGTG	789	[35]
	*sul*2	F: CGGCATCGTCAACATAACC R: GTGTGCGGATGAAAGTCAG	722	
	*sul*3	F: GAGCAAGATTTTTGGAATCG R: CATCTGCAGCTAACCTAGGGTTTGGA	792	
Tetracycline	*tet*A	F: GGCACCGAATGCGTATGAT R: AAGCGAGCGGGTTGAGAG	480	* [36]
	*tet*C	F: CTGGGCTGCTTCCTAATGC R: AGCTGTCCCTGATGGTCGT	580	
	*tet*M	F: GAGGTCCGTCTGAACTTTGCG R: AGAAAGGATTTGGCGGCACT	915	
	*tet*K	F: TTATGGTGGTTGTAGCTAGAAA R: AAAGGGTTAGAAACTCTTGAAA	382	[37]
	*tet*L	F: ATAAATTGTTTCGGGTCGGTAAT R: AACCAGCCAACTAATGACAATGAT	1077	[38]
Vancomycin	*van*A	F: GCAAGTCAGGTGAAGATGGA R: GCTAATACGATCAAGCGGTC	171	* [39]
	*van*B	F: GATGTGTCGGTAAAATCCGC R: CCACTTCGCCGACAATCAAA	271	
	*van*C1	F: GTATCAAGGAAACCTCGCGA R: CGTAGGATAACCCGACTTCC	836	
	*van*C2	F: GCAAACGTTGGTACCTGATG R: GGTGATTTTGGCGCTGATCA	523	
	*van*D	F: TGGAATCACAAAATCCGGCG R: TWCCCGCATTTTTCACAACS	311	
	*van*M	F: GGCAGAGATTGCCAACAACA R: AGGTAAACGAATCTGCCGCT	425	
	*van*N	F: CCTCAAATCAGCAGCTAGTG R: GCTCCTGATAAGTGATACCC	941	

* Multiplex PCR reaction. Thermocycler settings, including annealing, were applied following the respective reference indications. ARGs: Antibiotic resistance genes. Bp: base-pair. F: Forward. R: Reverse.

**Table 3 foods-12-01699-t003:** Identified bacterial strains in this research study.

Sources	Total Isolates	Gram+ and Gram−	Specimen Types
450 Fish food matrices	537 strains	331 Gram+ (61.6%)	80 mSD (24.1%)
			78 MSD (23.6%)
			73 mSP (22.1%)
			65 MSP (19.6%)
			15 mHPP (4.6%)
			20 MHPP (6.0%)
		206 Gram− (38.4%)	60 mSD (29.2%)
			57 MSD (27.6%)
			42 mSP (20.4%)
			37 MSP 18.0%)
			10 mHPP (4.8%)
			0 MHPP (0.0%)
100 Process samples	147 strains	94 Gram+ (63.9%)	62 OH (65.9%)
			32 S (34.1%)
		53 Gram− (36.1%)	31 OH (58.5%)
			22 S (41.5%)

m: *G. morhua*. M: *G. macrocephalus*. SD: Salted and Seasoned products. SP: Soaked Products HPP: High-Pressure Procedure treated products. OH: Operator hands. S: Surface samples.

**Table 4 foods-12-01699-t004:** Identified strains belonging to the *Staphylococcus* genus.

Staphylococci Strains		Sources							
	Tot.	mSD	mSP	mHPP	MSD	MSP	MHPP	OH	S
*S. aureus*	n. 51	n. 7 (13.7%)	n. 5 (9.8%)	-	n. 12 (23.5%)	n. 9 (17.6%)	-	n. 10 (19.6%)	n. 8 (15.8%)
*S. sciuri*	n. 88	n. 17 (19.3%)	n. 10 (11.4%)	n. 7 (7.9%)	n. 21 (23.9%)	n. 7 (7.9%)	n. 8 (9.1%)	n. 15 (17.1%)	n. 3 (3.4%)
*S. lentus*	n. 88	n. 15 (17.1%)	n. 21 (23.9%)	n. 5 (5.7%)	n. 14 (15.9%)	n. 13 (14.8%)	n. 7 (7.9%)	n. 11 (12.5%)	n. 2 (2.2%)
*S. saprophyticus*	n. 75	n. 20 (26.7%)	n. 12 (16.0%)	-	n. 10 13.3%	n. 18 (24.0%)	-	n. 3 (4.0%)	n. 12 (16.0%)
*S. xylosus*	n. 5	n. 2 (40.0%)	n. 2 (40.0%)	-	n. 1 (20.0%)	-	-	-	-
*S. haemolyticus*	n. 2	-	n. 2 (100.0%)	-	-	-	-	-	-
*S. simulans*	n. 1	-	n. 1 (100.0%)	-	-	-	-	-	-
*S. warneri*	n. 1	n. 1 (100.0%)	-	-	-	-	-	-	-

m: *G. morhua*. M: *G. macrocephalus*. SD: Salted and Seasoned products. SP: Soaked Products. HPP: High-Pressure Procedure-treated products. OH: Operator hands. S: Surface samples.

**Table 5 foods-12-01699-t005:** Identified Gram-negative strains in this research study.

Gram-Negative Strains	Sources								
	Tot.	mSD	mSP	mHPP	MSD	MSP	MHPP	OH	S
*E. coli*	n. 18	n. 8 (44.4%)	n. 5 (27.8%)	-	n. 3 (16.7%)	n. 2 (11.1%)	-	-	-
*Salmonella serovar.* Enteritidis	n. 1	n. 1 (100.0%)	-	-	-	-	-	-	-
*Cronobacter sakazakii*	n. 2	n. 1 (50.0%)	n.1 (50.0%)	-	-	-	-	-	-
*Acinetobacter lwofii*	n. 62	n. 13 (20.2%)	n. 6 (9.7%)	n. 1 (1.8%)	n. 17 (27.6%)	n. 15 (24.6%)	-	-	n. 10 (16.1%)
*Sphingomonas paucimobilis*	n. 50	n. 12 (24.0%)	-	n. 4 (8.0%)	n. 10 (20.0%)	n. 9 (18.0%)	-	n. 15 (30.0%)	-
*Pseudomonas luteola*	n. 41	n. 12 (29.3%)	n. 13 (31.7%)	-	n. 4 (9.7%)	-	-	n. 12 (29.3%)	-
*Serratia fonticola*	n. 37	-	n. 17 (45.9%)	-	n. 12 (32.4%)	n. 6 (16.3%)	-	-	n. 2 (5.4%)
*Pseudomonas fluorescens*	n. 27	n. 8 (29.7%)	-	n. 5 (18.5%)	n. 7 (25.9%)	-	-	-	n. 7 (25.9%)
*Citrobacter freundii*	n. 21	n. 5 (23.8%)	-	-	n. 4 (19.0%)	n. 5 (23.8%)	-	n. 4 (19.0%)	n. 3 (14.4%)

m: *G. morhua*. M: *G. macrocephalus*. SD: Salted and Seasoned products. SP: Soaked Products. HPP: High-Pressure Procedure treated products. OH: Operator hands. S: Surface samples.

**Table 6 foods-12-01699-t006:** Gram-positive bacterial strains: the phenotypic resistance patterns.

Gram-Positive Strains	Antibiotic Resistances (MIC * Values)					
	CLI	CTX	LNZ	OXA	TET	VAN
*S. aureus*	37/51 ** (≥4 µg/mL)	-	13/51 ** (≥8 µg/mL)	29/51 ** (≥4 µg/mL)	39/51 ** (≥16 µg/mL)	5/51 ** (≥32 µg/mL)
*S. sciuri*	65/88 ** (≥4 µg/mL)	3/88 ** (≥64 µg/mL)	10/88 ** (≥8 µg/mL)	52/88 ** (≥4 µg/mL)	73/88 ** (≥16 µg/mL)	15/88 ** (≥32 µg/mL)
*S. lentus*	37/88 ** (≥4 µg/mL)	-	7/88 ** (≥8 µg/mL)	69/88 ** (≥4 µg/mL)	77/88 ** (≥16 µg/mL)	13/88 ** (≥32 µg/mL)
*S. saprophyticus*	32/75 ** (≥4 µg/mL)	2/75 ** (≥64 µg/mL)	11/75 ** (≥8 µg/mL)	49/75 ** (≥4 µg/mL)	67/75 ** (≥16 µg/mL)	13/75 ** (≥32 µg/mL)
*S. xylosus*	1/5 (≥4 µg/mL)	-	-	-	2/5 (≥16 µg/mL)	-
*S. haemolyticus*	-	-	-	-	1/2 (≥16 µg/mL)	-
*E. faecalis*	23/44 ** (≥4 µg/mL)	-	27/44 ** (≥8 µg/mL)	-	35/44 ** (≥16 µg/mL)	17/44 ** (≥32 µg/mL)
*E. durans*	5/17 ** (≥4 µg/mL)	-	4/17 ** (≥8 µg/mL)	-	9/17 ** (≥16 µg/mL)	4/17 ** (≥32 µg/mL)
*E. faecium*	1/7 ** (≥4 µg/mL)	-	2/7 ** (≥8 µg/mL)	-	4/7 ** (≥16 µg/mL)	1/7 ** (≥32 µg/mL)
*K. kristinae*	-	-	-	-	5/24 (≥16 µg/mL)	-
*K. varians*	-	-	-	-	1/11 (≥16 µg/mL)	-

* MIC: Minimum Inhibitory Concentration. The obtained MIC values were compared to the CLSI Standard (CLSI, 2022). ** MDR: Multidrug resistant strains. CLI: Clindamycin, CTX: Cefotaxime; LNZ: Linezolid; OXA: Oxacillin; TET: Tetracycline; VAN: Vancomycin.

**Table 7 foods-12-01699-t007:** Gram-negative bacterial strains: the phenotypic resistance patterns.

Gram-Negative Strains	Antibiotic Resistances (MIC * Values)						
	AMK	CTX	ERP	CN	MRM	NIT	SUL
*E. coli*	-	6/18 ** (≥64 µg/mL)	2/18 ** (≥8 µg/mL)	-	-	2/18 ** (≥64 µg/mL)	-
*Salmonella* serovar Enteritidis	1/1 (≥64 µg/mL)	-	-	1/1 (≥16 µg/mL)	-	-	-
*C. sakazakii*	-	1/2 (≥64 µg/mL)	-	-	-	-	-
*A. lwofii*	8/62 (≥64 µg/mL)	-	-	4/62 (≥16 µg/mL)	-	-	-
*S. paucimobilis*	-	10/50 (≥64 µg/mL)	-	-	-	-	-
*P. luteola*	-	9/41 ** (≥64 µg/mL)	-	-	4/41 ** (≥16 µg/mL)	4/41 ** (≥512 µg/mL)	-
*S. fonticola*	-	4/37 (≥64 µg/mL)	-	-	-	3/37 (≥512 µg/mL)	-
*C. freundii*	-	1/21 (≥64 µg/mL)	-	-	-	-	1/21 (≥16 µg/mL)

* MIC: Minimum Inhibitory Concentration. The obtained MIC values were compared to the CLSI Standard [24]. ** MDR: Multidrug resistant strains. AMK: Amikacin; CTX: Cefotaxime; ERP: Ertapenem; CN: Gentamicin; MRM: Meropenem; NIT: Nitrofurantoin; SUL: Sulfamethoxazole.

**Table 8 foods-12-01699-t008:** ARGs detection from resistant and susceptible bacterial strains.

Bacterial Strain	Antibiotics	Resistant Strains	rARGs	Susceptible Strains	sARGs
*S. aureus*	AMK	-	-	51/51	12/51 *aac(3*)IV; 5/51 *aph*A1
	CLI	37/51	37/37 *erm*C; 19/37 *erm*B	14/51	4/14 *erm*C
	MRM; ERP	-	-	51/51	7/51 *bla*_KPC_; 3/51 *bla*_IMP_
	LNZ	13/51	13/13 *cfr*; 4/13 *optr*A; 1/13 *poxt*A	38/51	7/38 *cfr*; 10/38 *poxt*A
	OXA	29/51	29/29 *mec*A	22/51	11/22 *mec*A
	TET	39/51	39/39 *tet*C; 32/39 *tet*K, *tet*L	12/51	10/12 *tet*C; 2/12 *tet*A, *tet*B
	VAN	5/51	5/5 *van*D	-	-
	SUL	-	-	51/51	15/51 *sul*2; 8/51 *sul*3
*S. sciuri*	CLI	65/88	65/65 *erm*C; 42/65 *erm*B; 17/65*erm*A	23/88	9/23 *erm*C; 5/23 *erm*B
	CTX	3/88	3/3 *bla*_TEM_; 1/3 *bla*_CIT_	85/88	17/85 *bla*_TEM_; 5/85 *bla*_CTX-M_
	LNZ	10/88	10/10 *cfr*; 6/10 *poxt*A	78/88	21/78 *cfr*; 19/78 *optr*A
	OXA	52/88	52/52 *mec*A	36/88	19/36 *mec*A
	TET	73/88	73/73 *tet*C, *tet*K; 25/73 *tet*L	15/88	9/15 *tet*C, *tet*K, *tet*M
	VAN	15/88	15/15 *van*D; 2/15 vanN	73/88	4/73 *van*D
*S. lentus*	CLI	37/88	37/37 *erm*C; 12/37 *erm*B	51/88	13/51 *erm*C; 3/51 *erm*B
	LNZ	7/88	7/7 *cfr*; 1/7 *optr*A	81/88	6/81 *cfr*; 1/81 *optr*A
	OXA	69/88	69/69 *mec*A	19/88	5/19 *mec*A
	TET	77/88	77/88 *tet*C, *tet*K; 29/88 *tet*L	11/88	3/11 *tet*C, *tet*M
	VAN	13/88	13/13 *van*D	75/88	-
	SUL	-	-	88/88	4/88 *sul*2; 1/88 *sul*3
*S. saprophyticus*	CLI	32/75	32/32 *erm*C; 17/32 *erm*B	43/75	5/43 *erm*C
	CTX	2/75	2/2 *bla*_TEM_	73/75	-
	LNZ	11/75	11/11 *cfr*	64/75	-
	OXA	49/75	49/49 *mec*A	26/75	11/26 *mec*A
	TET	67/75	67/67 *tet*C, *tet*K; 41/67 *tet*L; 11/67 *tet*M	8/75	8/8 *tet*C; 3/8 *tet*M
	VAN	13/75	13/13 *van*D	62/75	-
*S. xylosus*	CLI	1/5	1/1 *erm*C	4/5	-
	TET	2/5	2/2 tetC, *tet*K, *tet*L	3/5	1/3 *tet*M
*S. haemolyticus*	TET	1/2	1/2 *tet*C	1/2	-
	AMK	-	-	44/44	5/44 *aph*A1
	CLI	23/44	23/23 *erm*C; 9/23 *erm*B	21/44	5/21 *erm*C
	LNZ	27/44	27/27 *cfr*; 17/27 *optr*A; 6/27 *poxt*A	17/44	9/17 *cfr*; 3/17 *optr*A
	SUL	-	-	44/44	2/44 *sul*2, *sul*3
	TET	35/44	35/35 *tet*C; 17/35 *tet*K, tetL; 5/35 *tet*M	9/44	3/9 *tet*C, *tet*M
	VAN	17/44	17/17 *van*D	27/44	-
*E. durans*	CLI	5/17	5/5 *erm*C	12/17	1/12 *erm*C
	LNZ	4/17	4/4 cfr; 2/4 *optr*A, *poxt*A	13/17	5/13 *cfr*
	TET	9/17	9/9 *tet*C, *tet*K, *tet*L, *tet*M	8/17	6/8 *tet*C, *tet*K
	VAN	4/17	4/4 *van*D	13/17	-
*E. faecium*	CLI	1/7	1/1 *erm*C	6/7	-
	LNZ	2/7	2/2 *cfr*, *optr*A; 1/2 *poxt*A	5/7	-
	TET	4/7	4/4 *tet*C, *tet*L; 1/4 *tet*M	3/7	3/3 *tet*C, *tet*M
	VAN	1/7	1/1 *van*D	6/7	-
*K. kristinae*	TET	5/24	5/5 *tet*C; 1/5 *tet*K, *tet*L	19/24	7/19 *tet*C
	MRM	-	-	24/24	3/24 *bla*_NDM_
*K. varians*	TET		1/1 *tet*C		4/11 *tet*K, *tet*L
	CTX	6/18	6/6 *bla*_TEM_, *bla*_CTX-M_	12/18	7/18 *bla*_CIT_, *bla*_TEM_, *bla*_CTX-M_
	ERP	2/18	2/2 *bla*_IMP_; *bla*_KPC_; 1/2 *bla*_OXA-48-like_	16/18	5/16 *bla*_KPC_; 3/16 *bla*_IMP_
	NIT	2/18	2/2 *nfs*A, *nfs*B	16/18	-
	SUL	-	-	18/18	4/18 *sul*2
*Salmonella serovar.* Enteritidis	AMK	1/1	1/1 *aphA*1	-	
	CN	1/1	-	-	
	MRM	-	-	1/1	1/1 *bla*_OXA-48-like_;
	NIT	-	-	1/1	1/1 *nfs*A, *nfs*B;
	SUL	-	-	1/1	1/1 *sul*2;
	TET	-	-	1/1	1/1 *tet*A, *tet*C, *tet*M
*C. sakazakii*	CTX	1/2	1/2 *bla*_TEM_	1/2	-
	ERP	-	-	-	1/2 *bla*_IMP_
*A. lwofii*	AMK	8/62	8/8 *aph*A1; 3/8 *aac(3)*IV	54/62	-
	CN	4/62		58/62	-
	CTX	-	-	62/62	4/62 *bla*_CIT_, *bla*_CTX-M_
*S. paucimobilis*	CTX	10/50	10/10 *bla*_TEM_; 4/10 *bla*_CIT_	40/50	-
	ERP	-	-	50/50	7/50 *bla*_KPC_, *bla*_IMP_
	SUL	-	-	50/50	9/50 *sul*2
*P. luteola*	CTX	9/41	9/9 *bla*_TEM_, *bla*_CIT_; 2/9 *bla*_CTX-M_	32/41	-
	MRM	4/41	4/4 *bla*_IMP_, *bla*_OXA-48-like_	37/41	-
	NIT	4/41	4/4 *nfs*A, *nfs*B	37/41	-
*S. fonticola*	CTX	4/37	4/4 *bla*_TEM_; 1/4 *bla*_CIT_	33/37	5/33 *bla*_CTX-M_
	MRM	-	-	37/37	7/37 *bla*_IMP_
	NIT	3/37	3/3 *nfs*A, *nfs*B	34/37	-
	SUL			37/37	9/37 *sul*2, *sul*3
*C. freundii*	CTX	1/21	1/21 *bla*_TEM_	20/21	-
	SUL	1/21	1/21 *sul*2, *sul*3	20/21	8/20 *sul*2

rARGs: target genes isolated from resistant strains. sARGs: target genes isolated from susceptible strains.

## Data Availability

Data is contained within the article.

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
