# Peer review of "Antibiotic Resistance Profiles and ARG Detection from Isolated Bacteria in a Culture-Dependent Study at the Codfish Industry Level"

_foods, 2023, doi:10.3390/foods12081699_

Round 1

Reviewer 1 Report

1、The conclusion is lengthy and the description of the exprimental conclusion is simplified.

2、The p should be italic, please modify this issue in the manuscript.

3、Line 310, (55.88% - 95% CI:44.08%-67.68%). The spaces in the manuscript are different and appear multiple times. Please check the entire text.

4、The amount of experiments in the manuscript is small, and some exprimental indicators can be added to further prove your viewpoint.

5、The discussion section in the manuscipt should be more in-depth.

Author Response

Dear REVIEWER 1,

We are grateful to considerer our manuscript entitled “Antibiotic resistance profiles and ARGs detection from isolated bacteria in a culture-dependent study at codfish industry level” to be considered for publication in the FOODS Journal in the Special Issue “Occurrence and Control of Antibiotic Resistant Strains of Bacteria in Food Chain”.

All Reviewers’ suggestions resulted precious and important for the improving of its scientific quality.

In the following paragraphs, we reported all changes.

1、The conclusion is lengthy and the description of the exprimental conclusion is simplified.

Authors (lines 586-605): The conclusions section has been revised.

2 The p should be italic, please modify this issue in the manuscript.

Authors (line 353): modified.

Authors (line 356): modified.

Authors (line 372): modified.

Authors (line 384): modified.

Authors (line 385): modified.

Authors (line 386): modified.

Authors (line 388): modified.

Authors (line 390): modified.

Authors (line 393): modified.

Authors (line 394): modified.

Authors (line 395): modified.

Authors (line 396): modified.

Authors (line 398): modified.

Authors (line 399): modified.

Authors (line 455): modified.

Authors (line 522): modified.

3、Line 310, (55.88% - 95% CI:44.08%-67.68%). The spaces in the manuscript are different and appear multiple times. Please check the entire text.

Authors (lines 268-269): corrected.

Authors (lines 281-282): corrected.

Authors (lines 301-303): corrected.

Authors (lines 309-312): corrected.

Authors (lines 323-324): corrected.

Authors (lines 390-393): corrected.

Authors (lines 483-485): corrected.

Authors (lines 492-494): corrected.

4、The amount of experiments in the manuscript is small, and some exprimental indicators can be added to further prove your viewpoint.

Authors: The present manuscript is part of a PhD project, as preliminary biomolecular investigation focused on the AMR phenomenon, performed in isolated bacteria from food matrices and environments at food industry level. Therefore, the purposed analytical parameters, for qualitative screenings, result conform for the first scientific aims.

5、The discussion section in the manuscipt should be more in-depth.

Authors (lines 540-575): This part has been improved.

We confirm that neither the manuscript nor any parts of its content are currently under consideration or published in another journal.

All authors have approved the manuscript and agree with its submission to the FOODS Journal.

We appreciate the possibility to publish our paper and believe that our manuscript will be of interest to You and to the readers of Your journal.

Thank You for Your time and attention.

Best regards,

Gianluigi Ferri

Doctor in Veterinary Medicine (D.V.M.)

Ph.D. Student in Food Inspection

Faculty of Veterinary Medicine; University of Teramo, Italy.

Reviewer 2 Report

Reviewer 1

Reviewers Comments

In the present manuscript, the authors aimed to identify bacteria, from food producing processes and industrial environments, bacteria were screened for phenotypic and genotypic resistance determinants. Authors also isolated a total of 684 bacterial strains. Authors also aimed antibiotic susceptibility tests that showed resistances against tetracycline, oxacillin, and clindamycin in the Staphylococcus genus (both from food and environmental samples), against beta-lactams (cefotaxime, carbapenems, etc.) and nitrofurans (nitrofurantoin) from E. coli and Salmonella enterica serovar. Authors claim that this study determined high antibiotic resistance genes circulation in a fish food industrial chain from macro to microenvironment. Authors also claim that the obtained data confirmed the “antibiotic resistance phenomenon" diffusion, and its repercussions on the health and food-producing systems.

Question 1: There are some typing and spacing errors in the manuscript i.e., in table 1 last line.

Question 2: I don’t think selection of pathogen on the basis of selective media is a reliable method. Why not authors are switching towards 16s and 18s RNA sequencing after initial screening?

Question 3: Authors have provided a detailed list of F/R primers for target gene, Authors should also add gel image based on PCR product, which will further strengthen your results.

Question 4: Why the authors did not perform any biochemical characterization for the identified strains?

Questions 5: Authors should clearly summarize the resistance of bacterial strains to antibiotics, and which antibiotic is best suitable for the said strains should be clealy mentioned for the reader of Foods.

English editing of the text must be done as some sentence must be rewritten and typos should be corrected.

Author Response

Dear REVIEWER 2,

We are grateful to considerer our manuscript entitled “Antibiotic resistance profiles and ARGs detection from isolated bacteria in a culture-dependent study at codfish industry level” to be considered for publication in the FOODS Journal in the Special Issue “Occurrence and Control of Antibiotic Resistant Strains of Bacteria in Food Chain”.

All Reviewers’ suggestions resulted precious and important for the improving of its scientific quality.

In the following paragraphs, we reported all changes.

Question 1: There are some typing and spacing errors in the manuscript i.e., in table 1 last line.

Authors: All manuscript has been revised.

Question 2: I don’t think selection of pathogen on the basis of selective media is a reliable method. Why not authors are switching towards 16s and 18s RNA sequencing after initial screening?

Authors: The selective cultures, successively associated with the biochemical automated identification method (VITEK® 2 system, bioMérieux, Paris, France), was confirmed using the mass spectrometry MALDI-TOF (Matrix Assisted Laser Desorption Ionization – Time of Flight) as accurate method.

Question 3: Authors have provided a detailed list of F/R primers for target gene, Authors should also add gel image based on PCR product, which will further strengthen your results.

Authors (lines 388-393): Gel image introduced.

Authors (lines 408-412): Gel image introduced.

Question 4: Why the authors did not perform any biochemical characterization for the identified strains?

Authors (lines 168-175): This part has been improved.

Questions 5: Authors should clearly summarize the resistance of bacterial strains to antibiotics, and which antibiotic is best suitable for the said strains should be clealy mentioned for the reader of Foods.

Authors (lines 344-349): This section has been improved.

Concerning the best suitable antibiotic for the said strains, it was not considered because the present study aims to mainly focus on the resistance phenotypic and genotypic obtained patterns.

We confirm that neither the manuscript nor any parts of its content are currently under consideration or published in another journal.

All authors have approved the manuscript and agree with its submission to the FOODS Journal.

We appreciate the possibility to publish our paper and believe that our manuscript will be of interest to You and to the readers of Your journal.

Thank You for Your time and attention.

Best regards,

Gianluigi Ferri

Doctor in Veterinary Medicine (D.V.M.)

Ph.D. Student in Food Inspection

Faculty of Veterinary Medicine; University of Teramo, Italy.

Reviewer 3 Report

The manuscript titled “Antibiotic resistance profiles and ARGs detection from isolated 2 bacteria in a culture-dependent study at codfish industry level” is important and the authors have comprehensively elucidated the presence of ARGs in culture based studies at codfish industry. Of course, ARGs are potential concerns in various food industries, and the study has impact. However, the present study has substantial concerns that needs to be addressed before further consideration.  The latest information in the literature is missing in the background, in fact what are the major factors that contribute the presence of the accumulation of ARGs in codfish industry needs to be illustrated. Also, the gap evident in the literature and how the present study mitigates the gaps are also important. The objectives are not clear and that should be specifically described in eth last section of the introduction.

The quantitative study on the sample collection provides little information, it is important to clearly specify the methods that can be used to reproduce similar study by other researchers.

The microbiological details for the qualitative screenings seem to be incomplete.  

The concentration of the antibiotics is not clear in the present study.

Which version of the CLSI standards are used for the present study? There are no such references for such version??

The authors have not tried Vancomycin, the present generation drug for Staphylococcus aureus??Infact, methicillin is not effective in several cases, it showed extensive resistance.

What are the major primers used for the amplification of the genes? In table 2, nucleotide sequence means primers? It should be properly mentioned over there.

What are the comparative controls used for the experiments? Not clear.

The results are not described based on the tables and figures propel in several places.

The uniqueness and novelty of the present study should be highlighted in the conclusion section.

The manuscript requires professional proof verification and editing. The overall presentation and language need substantial modification and revision.

The manuscript requires professional proof verification and editing. The overall presentation and language needs substantial modification and revision.

Author Response

Dear REVIEWER 3,

We are grateful to considerer our manuscript entitled “Antibiotic resistance profiles and ARGs detection from isolated bacteria in a culture-dependent study at codfish industry level” to be considered for publication in the FOODS Journal in the Special Issue “Occurrence and Control of Antibiotic Resistant Strains of Bacteria in Food Chain”.

All Reviewers’ suggestions resulted precious and important for the improving of its scientific quality.

In the following paragraphs, we reported all changes.

The quantitative study on the sample collection provides little information, it is important to clearly specify the methods that can be used to reproduce similar study by other researchers.

The microbiological details for the qualitative screenings seem to be incomplete. 

Authors: All qualitative microbiological procedures have been performed following the International Standards. It is our opinion that a further specification of laboratory processes could be unusual if compared with the manuscript’s aims. Therefore, we suggest maintaining the current form.

The concentration of the antibiotics is not clear in the present study.

Authors: The specific antibiotic concentrations were not introduced because the used instruments VITEK® 2 system (bioMérieux, Paris, France), for the antibiotic susceptibility tests, were tested at different antibiotic concentrations.

In agreement with the manufacturer instructions, AST-N379; AST-P658 and AST-P659 (bioMérieux, Paris, France) molecules’ concentration are available at the following links:

  • https://www.biomerieux.hu/sites/subsidiary_hu/files/nb_publications-vitek_2_9308339008gba_web.pdf;
  • https://www.biomerieux-usa.com/sites/subsidiary_us/files/prn_052570_rev_03.a_cln_idast_vitek_2_card_guide_final_art.pdf;
  • https://www.biomerieux-usa.com/sites/subsidiary_us/files/prn_052571_rev_02.a_cln_idast_vitek2-cards-brochure_final_art_2.pdf.

Which version of the CLSI standards are used for the present study? There are no such references for such version??

Authors (lines 207-208): Added.

The authors have not tried Vancomycin, the present generation drug for Staphylococcus aureus??Infact, methicillin is not effective in several cases, it showed extensive resistance.

Authors (line 205): Vancomycin was considered in the antibiotic susceptibility tests. Indeed, this molecule is normally included both in the AST-P658 and AST-P658 cards. Therefore, phenotypical and genotypical resistance patterns were studied.

What are the major primers used for the amplification of the genes? In table 2, nucleotide sequence means primers? It should be properly mentioned over there.

Authors (line 234): Table 2 revised.

What are the comparative controls used for the experiments? Not clear.

Authors: Negative controls and relative positive ones were included in all assays, as graphically illustrated by the introduced Figures 2 and 3.

The results are not described based on the tables and figures propel in several places.

Authors (lines 272-350): Improved sections.

Authors (lines 394-425): Improved section.

The uniqueness and novelty of the present study should be highlighted in the conclusion section.

Authors (lines 824-844): Conclusions section has been improved.

The manuscript requires professional proof verification and editing. The overall presentation and language needs substantial modification and revision.

Authors: substantial revisions have been performed.

We confirm that neither the manuscript nor any parts of its content are currently under consideration or published in another journal.

All authors have approved the manuscript and agree with its submission to the FOODS Journal.

We appreciate the possibility to publish our paper and believe that our manuscript will be of interest to You and to the readers of Your journal.

Thank You for Your time and attention.

Best regards,

Gianluigi Ferri

Doctor in Veterinary Medicine (D.V.M.)

Ph.D. Student in Food Inspection

Faculty of Veterinary Medicine; University of Teramo, Italy.

Reviewer 4 Report

The manuscript, foods-2356375, provides antibiotic resistance profiles as phenotypic expression of resistance genes amplified from isolated bacterial strains derived from differentially processed codfish products and industrial environments. I think this is valuable work. The manuscript is well carried out, well written, and the data obtained adequately support the conclusions. In addition, the results are well compared with the data available in the literature.

The English language is acceptable.

I suggest acceptance of the manuscript in its present form.

Author Response

Dear REVIEWER 4,

We are grateful to considerer our manuscript entitled “Antibiotic resistance profiles and ARGs detection from isolated bacteria in a culture-dependent study at codfish industry level” to be considered for publication in the FOODS Journal in the Special Issue “Occurrence and Control of Antibiotic Resistant Strains of Bacteria in Food Chain”.

All Reviewers’ suggestions resulted precious and important for the improving of its scientific quality.

I suggest acceptance of the manuscript in its present form.

Authors: We really appreciate Reviewer’s observations.

We confirm that neither the manuscript nor any parts of its content are currently under consideration or published in another journal.

All authors have approved the manuscript and agree with its submission to the FOODS Journal.

We appreciate the possibility to publish our paper and believe that our manuscript will be of interest to You and to the readers of Your journal.

Thank You for Your time and attention.

Best regards,

Gianluigi Ferri

Doctor in Veterinary Medicine (D.V.M.)

Ph.D. Student in Food Inspection

Faculty of Veterinary Medicine; University of Teramo, Italy.

Round 2

Reviewer 2 Report

The R1 is more logical and the new figures added are clear and supporting the conclusion. Almost all of my queries are answered bu authors. Thus I would like to recommend this manuscript for publication in Foods.

Author Response

Dear REVIEWER 2,

We are grateful to considerer our manuscript entitled “Antibiotic resistance profiles and ARGs detection from isolated bacteria in a culture-dependent study at codfish industry level” to be considered for publication in the FOODS Journal in the Special Issue “Occurrence and Control of Antibiotic Resistant Strains of Bacteria in Food Chain”. All Reviewers’ suggestions resulted precious and important for the improving of its scientific quality.

Reviewer 2

The R1 is more logical and the new figures added are clear and supporting the conclusion. Almost all of my queries are answered bu authors. Thus I would like to recommend this manuscript for publication in Foods.

Authors: We really appreciate Reviewer’s observations.

This paper involves different scientific disciplines: food microbiology, food safety and technology. Therefore, we believe that the presented manuscript will be able to fit with Your Journal aims and scopes.

We confirm that neither the manuscript nor any parts of its content are currently under consideration or published in another journal.

All authors have approved the manuscript and agree with its submission to the FOODS Journal.

We appreciate the possibility to publish our paper and believe that our manuscript will be of interest to You and to the readers of Your journal.

Thank You for Your time and attention.

Best regards,

Gianluigi Ferri

Doctor in Veterinary Medicine (D.V.M.)

Ph.D. Student in Food Inspection

Faculty of Veterinary Medicine; University of Teramo, Italy.

Reviewer 3 Report

The manuscript has been improved. But there are still some comments that are not addressed in the revision needs to be addressed before further consideration.

Mentioned.

Author Response

Dear REVIEWER 3,

We are grateful to considerer our manuscript entitled “Antibiotic resistance profiles and ARGs detection from isolated bacteria in a culture-dependent study at codfish industry level” to be considered for publication in the FOODS Journal in the Special Issue “Occurrence and Control of Antibiotic Resistant Strains of Bacteria in Food Chain”. All Reviewers’ suggestions resulted precious and important for the improving of its scientific quality.

Reviewer 3

The manuscript has been improved. But there are still some comments that are not addressed in the revision needs to be addressed before further consideration.

Authors: Manuscript has been generally improved.

Authors (lines 152-155): Improved section.

Authors (lines 161-163): Improved section.

This paper involves different scientific disciplines: food microbiology, food safety and technology. Therefore, we believe that the presented manuscript will be able to fit with Your Journal aims and scopes.

We confirm that neither the manuscript nor any parts of its content are currently under consideration or published in another journal.

All authors have approved the manuscript and agree with its submission to the FOODS Journal.

We appreciate the possibility to publish our paper and believe that our manuscript will be of interest to You and to the readers of Your journal.

Thank You for Your time and attention.

Best regards,

Gianluigi Ferri

Doctor in Veterinary Medicine (D.V.M.)

Ph.D. Student in Food Inspection

Faculty of Veterinary Medicine; University of Teramo, Italy.
